# Strict molecular sieving over electrodeposited 2D-interspacing-narrowed graphene oxide membranes

Benyu Qi[1,2], Xiaofan He[1,3], Gaofeng Zeng [1], Yichang Pan[4], Guihua Li[1], Guojuan Liu[1], Yanfeng Zhang[1,3], Wei Chen [1] & Yuhan Sun[1,3]

To separate small molecules/species, it's crucial but still challenging to narrow the 2D-interspacing of graphene oxide (GO) membranes without damaging the membrane. Here the fast deposition of ultrathin, defect-free and robust GO layers is realized on porous stainless steel hollow fibers (PSSHFs) by a facile and practical electrophoresis deposition (ED) method. In this approach, oxygen-containing groups of GO are selectively reduced, leading to a controlled decrease of the 2D channels of stacked GO layers. The resultant ED-GO@PSSHF composite membranes featured a sharp cutoff between C2 (ethane and ethene) and C3 (propane and propene) hydrocarbons and exhibited nearly complete rejections for the smallest alcohol and ion in aqueous solutions. This demonstrates the versatility of GO based membranes for the precise separation of various types of mixtures. At the same time, a robust mechanical strength of the ED-GO@PSSHF membrane is also achieved due to the enhanced interaction at GO/support and GO/GO interfaces.

[1] CAS Key Laboratory of Low-carbon Conversion Science and Engineering, Shanghai Advanced Research Institute, Chinese Academy of Sciences, Shanghai 201210, China. [2] University of Chinese Academy of Sciences, Beijing 100049, China. [3] School of Physical Science and Technology, ShanghaiTech University, Shanghai 201210, China. [4] State Key Laboratory of Materials-Oriented Chemical Engineering, College of Chemical Engineering, Nanjing Tech University, Nanjing 211800, China. Benyu Qi and Xiaofan He contributed equally to this work. Correspondence and requests for materials should be addressed to G.Z. (email: zenggf@sari.ac.cn) or to Y.S. (email: sunyh@sari.ac.cn)

Graphene oxide (GO), an atomically thin 2D structure encompassing oxygen-containing groups, has been widely considered for membrane separation applications because the 2D channels of stacked GO provide molecular sieving capabilities[1–4]. Benefiting from water preferential adsorption and fast capillary diffusion[1], GO membranes have exhibited excellent water permselectivity for various aqueous mixtures like water/solvents[5–9], dyed water[10], and saline water[11–13]. Moreover, high selectivities for the separation of organics were also obtained through the exclusion of large molecules[8, 9, 14]. In the case of gas separation, exciting selectivities of $H_2/N_2$, $H_2/CO_2$ and $CO_2/N_2$ have been achieved through elaborate controls of GO membrane preparation[3, 4, 15]. Actually, GO is a collective name for exfoliated graphitic oxygenates with diverse oxidation degrees[16]. Thus, GO membranes can be tailored for these various separations because the oxidation degree determines the size of 2D channels of stacked GO. To cater to the specific separations, therefore, it's crucial but still challenging to control the 2D channel size of GO membranes.

The channel could be somewhat enlarged by inserting hetero-species[8, 10, 17], which endow GO membranes with large-size species separation capacity. On the contrary, narrowing the 2D interspacing of GO-based membranes is more attractive as the tighter GO membranes are actually desired for small molecule sieving[13, 17–19]. Theoretically, the 2D channel of stacked GO could be narrowed near to zero through different reduction levels[20]. Various reduction methods, like chemical reduction and thermal deoxygenation, have been purposed for the preparation of reduced graphene oxide (rGO)[20, 21]. In the case of supported membrane preparation, however, GO reduction either before or after membrane formation is always incompatible with achieving perfect membranes[22]. The pre-reduction of GO weakens its dispersibility and increases disorder in the membrane, while the post-reduction of supported GO membranes normally leads to significant deformations and defects. As an alternative, in situ GO reduction during membrane formation is greatly expected to narrow GO 2D channels without damaging the membranes.

Electrophoresis deposition (ED) is a well-developed and economical technology where charged colloids in suspension migrate and deposit onto the surface of an electrode. For thin-film fabrication, it can offer advantages of high deposition rate, thickness controllability, good uniformity, strong adhesion and facile scale-up[23, 24]. The ED method is well suited for preparation of GO thin films because the highly hydrophilic and easily deprotonated functional groups endow GO hydrosol with uniform dispersibility and negative charges[25]. In this way, GO thin films have been prepared for electrochemistry, anti-corrosion, and optics applications[24, 26, 27]. Importantly, GO could be partially deoxygenated in the ED process[25, 26, 28]. Therefore, ED is a promising method to prepare uniform GO membranes with narrower 2D channels. To address the issues of mechanical strength and practicality of soft GO membranes[29], moreover, porous stainless steel hollow fiber (PSSHF) is a superior GO support owing to its rigidity, high surface-to-volume ratio and packing density, facile scale-up and affordability. In addition, PSSHF can be directly used as a working electrode for membrane deposition due to its conductivity.

Here the deposition of ultrathin and defect-free GO layers was realized on PSSHFs through an ED process. The 2D channels of the resultant GO membranes are well narrowed and controlled. The obtained ED-GO@PSSHF composite membranes exhibit excellent separation performance for C2/C3 hydrocarbons, alcohol dehydration and desalination as well as enhanced stability.

## Results

**Electrophoresis deposition of GO@PSSHF membranes.** Figure 1a illustrates the anodic ED process employed for the fabrication of the GO@PSSHF membrane. A circular electric field was formed between the two concentric electrodes of PSSHF

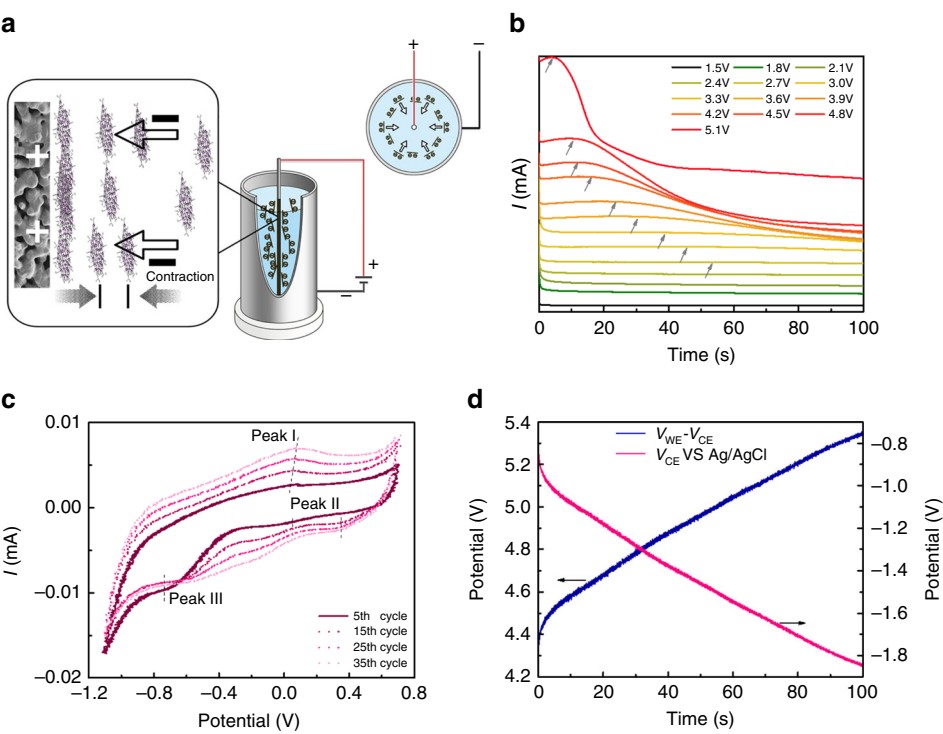

**Fig. 1** Electrophoresis deposition of graphene oxide on PSSHF. **a** Schematic of GO electrophoresis deposition on PSSHF with a circular electric field, **b** current variation with fixed DC voltages of $V_{WE}$–$V_{CE}$ in the GO electrophoresis deposition, **c** CV curves of GO on the glass-carbon electrode at room temperature and **d** time dependence of voltages $V_{WE}$–$V_{CE}$ and $V_{CE}$ in the three-electrode system for GO deposition with a $V_{WE}$ of 3.2 V

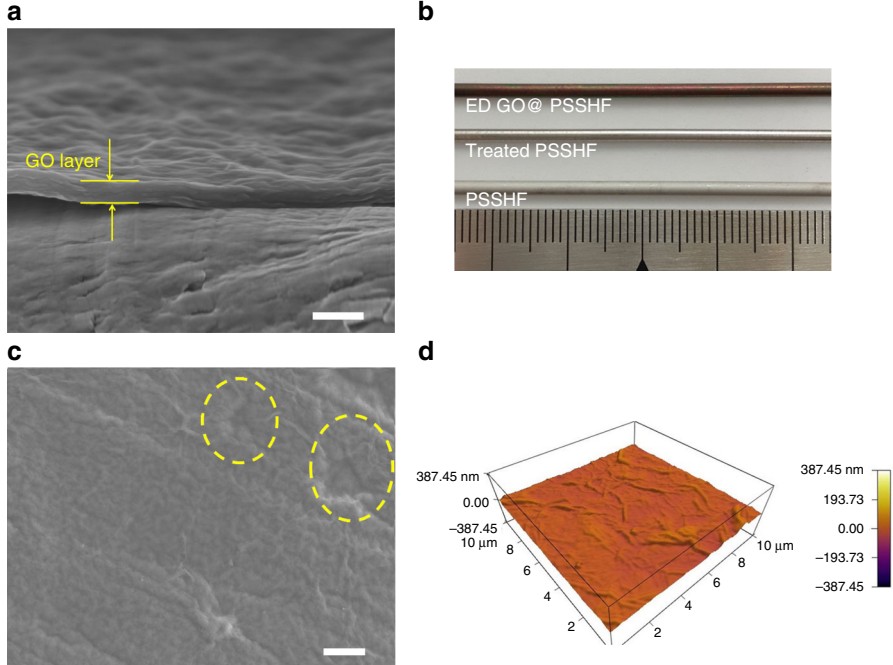

**Fig. 2** Morphology of ED-GO@PSSHF membrane. SEM images of **a** cross-section and **c** surface of ED-GO@PSSHF membrane, **b** optical image of PSSHFs and ED-GO@PSSHF, **d** atomic force microscopy (AFM) measurement of ED-GO surface, scale bars **a** 200 nm and **c** 2 μm

(anode/working electrode, WE) and stainless steel tubular container for the GO suspension (cathode/counter electrode, CE), which insured a uniform driving force from every direction (Supplementary Figs. 1 and 2). Applying a DC voltage, the negative GO colloids were partially reduced by the cathode, and then driven by the electrostatic force to stack onto the surface of the PSSHF[30]. To optimize the ED conditions, the effects of voltage $V_{WE}-V_{CE}$, electrode spacing and working time on membrane thickness and GO composition were investigated. At constant $V_{WE}-V_{CE}$, the current density reflects the migration and deposition rate of GO colloids. As shown in Fig. 1b, the current density depends on $V_{WE}-V_{CE}$, which remained first stable on a plateau and then declined with time. In the range of 4.0–4.8 V, the I–t curves show clear and relative long plateaus of current density. It's worth noting that low current density leads to slow deposition and inhomogeneous coverage whereas high current density results in fast deposition but disorder of GO on the support. In addition, fine bubbles, probably consisting of oxygen from water electrolysis and/or $CO_2$ from deoxidization of functional groups[28], are generated on the PSSHF at high $V_{WE}-V_{CE}$, which could damage the membrane quality. As shown in Fig. 1c, the cyclic voltammogram (CV) at the initial scan yields a reduction signal (peak III) at −0.77 V which can be assigned to the irreversible reduction of GO, indicating that GO could be reduced if $V_{CE}$ is more negative than that value. On the other hand, the stable oxidation peak I and reduction peak II in the successive scans are ascribed to the redox pair of some active oxygen-containing groups of GO that cannot be reduced by the CV tests, suggesting a selective and partial reduction of GO in the ED process[26]. In a three-electrode system, $V_{CE}$ was more negative than −0.77 V when $V_{WE}-V_{CE}$ was varied in the range of 4.3–5.3 V (Fig. 1d). It indicates that GO reduction will occur when the DC voltage of $V_{WE}-V_{CE}$ was set higher than 4.3 V. However, the ED-GO layer quality was difficult to control under high $V_{WE}-V_{CE}$ due to the very fast deposition rate and impact of bubbles (Supplementary Fig. 3). Thus the $V_{WE}-V_{CE}$ was normally set around 4.5 V. In this ED conditions, the thickness of ED-GO@PSSHF exhibits nearly linear dependent on the ED time

within 90 s (Supplementary Fig. 4). The C/O ratio of ED-GO layer increased fast from 2.2 to 2.6 in the beginning and then sluggishly grew to around 2.7 in the following. At the same time, a depth profile analysis of the ED-GO membrane reveals that the C/O ratio increased from the GO/PSSHF interface but remained relatively stable in/on the membrane bulk and top surface (Supplementary Fig. 5). In addition, the deposition rate was evidently impacted by the electrode gap (Supplementary Fig. 6). Therefore, taking GO reduction, bubble impact, thickness and defects control into account, the ED-GO@PSSHF membranes described in the following were fabricated in a 1 mg mL$^{-1}$ GO suspension with 9 mm electrode spacing and 4.5 V $V_{WE}-V_{CE}$ during 35 s.

**Morphology and structure of ED-GO@PSSHF membranes.** The scanning electron microscope (SEM) image of the membrane cross-section shows a uniform thickness of 95 nm with a highly ordered and well-packed 2D lamellar structure, indicating that GO flakes are tightly stacked in the in-plane direction (Fig. 2a). The ED-GO@PSSHF membrane had a homogenously brown color (Fig. 2b), slightly darker than the typical GO membranes shown in Supplementary Fig. 7 and literature[1, 7]. The surface view of the membrane further reveals a dense coverage without visible defects, with large defects of the PSSHF (dash lines) even being completely covered (Fig. 2c). It also shows typical wrinkles, which were probably caused by the stacked GO boundaries, in SEM and AFM images (Figs. 2c and d). These wrinkles are helpful for the membrane's permeability and the mechanical stability because wrinkles can act not only as entrances but also buffering spaces upon the intercalation of permeation species[3].

The structures of pristine GO and ED-GO were analyzed by XRD (Fig. 3a). The pristine GO exhibited the intense peak of graphite oxide (001) at 2Θ = 11.1° (d-value = 0.80 nm) without any signal of a graphitic structure. However, the peak of ED-GO (001) was slightly up-shifted to 12.1°, translating to a smaller d-value of 0.73 nm. Deducting the single layer thickness of graphite or well-reduced GO (ca. 0.34–0.40 nm)[1, 2], the average

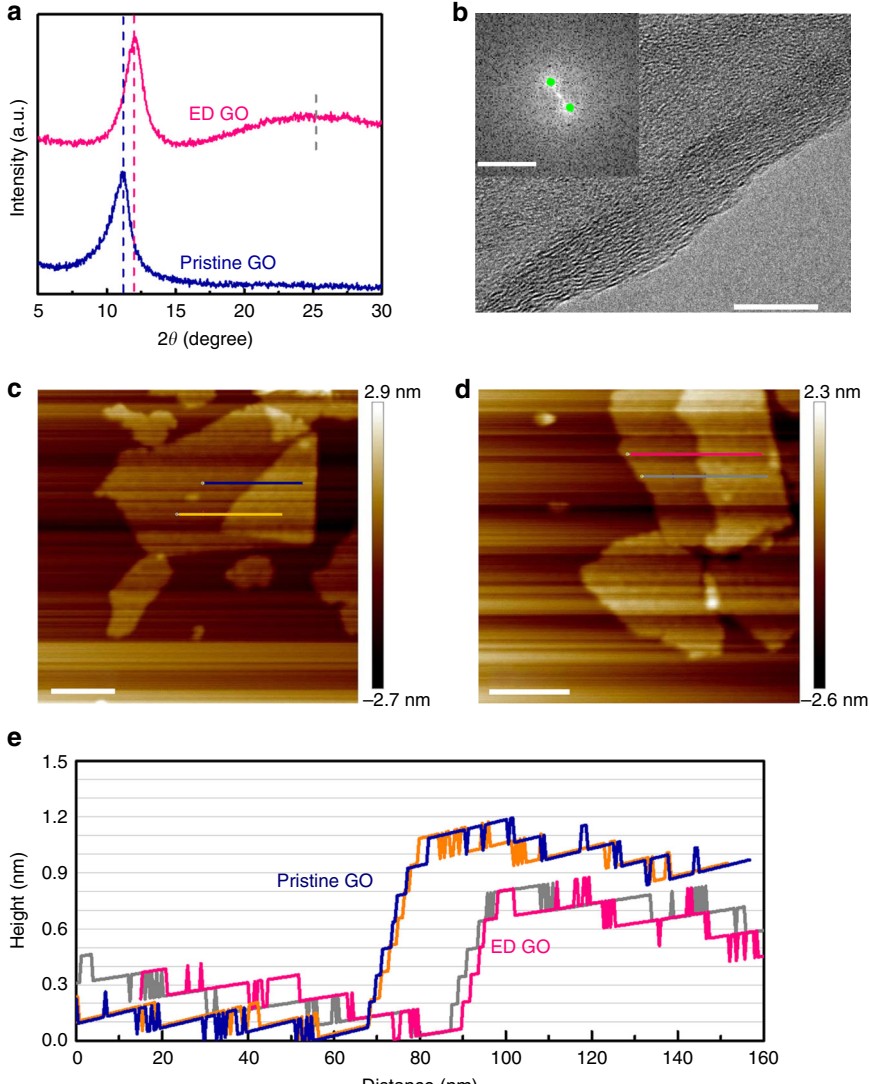

**Fig. 3** Structure analysis of ED-GO layers. **a** X-ray diffraction (XRD) patterns of pristine and ED-GO, **b** HRTEM cross-section image of ED-GO fragment (inset: fast Fourier transform of TEM image), AFM measurements of **c** pristine GO and **d** ED-GO bilayers and **e** layer height profiles of the GO bilayers (offsets made for clarity), scale bars **b** 10 nm, inset **b** 5 nm$^{-1}$ and **c**, **d** 100 nm

2D channel size of ED-GO is estimated at $0.36 \pm 0.03$ nm. In contrast, the channel size of pristine GO was $0.43 \pm 0.03$ nm. In addition, a weak signal of graphite (002) was observed at 25.5°[23]. It reveals that the ED-GO layer was slightly reduced, which indicated a successful narrowing of the 2D channel. In comparison with the broadened (001) peak of pristine GO, ED-GO exhibits a sharp and symmetrical (001) peak, indicating that the ED-GO possesses a uniform orientation and stacking structure. The high resolution transmission electron microscope (HRTEM) cross-section view of the ED-GO fragment depicts a highly ordered GO morphology (Fig. 3b). From the fast Fourier transform (FFT) of Fig. 3b, the interplanar spacing of the stacked ED-GO was derived as $0.70 \pm 0.02$ nm by a reciprocal translating with $2/d$ nm$^{-1}$, where d nm$^{-1}$ is the diffraction spots distance (Fig. 3b inset). It is fairly close to the $d$-value determined by XRD. From the AFM analyses of GO bilayers, the thickness of pristine GO monolayer is averaged at 1.02 nm, in line with the literature[18], while the ED-GO monolayer thickness is ~0.76 nm (Figs. 3c–e). Deducting the graphite or rGO monolayer, the 2D channel size between the ED-GO bilayer is $0.39 \pm 0.03$ nm, a little bit larger but very close to that estimated from the diffraction result. Thus the narrowed 2D channel of ED-GO is further confirmed by the bilayer measurements. The height profile of ED-GO layer exhibits a smooth surface with reasonable roughness as well as a continuous surface texture without visible defects, implying that the ED process has no appreciable impact on the surface structure of GO (Fig. 3e). Therefore, the quantitative XRD analysis with support of the semi-quantitative results from TEM and AFM leads to the conclusion that the 2D channels of ED-GO were narrowed during the ED process.

**Chemical properties of ED-GO@PSSHF membranes**. To identify the chemical states of ED-GO, Fourier transform infrared (FTIR) spectroscopy, X-ray photoelectron spectroscopy (XPS) and Raman spectroscopy were employed to analyze the pristine GO and ED-GO layers. In the FTIR spectrum of pristine GO (Fig. 4a), the broad peak centered at 3250 cm$^{-1}$ is attributed to the stretching vibrations of C-OH and the intercalated H$_2$O. The H-bonds between hydroxyl and carboxyl groups probably lead to peak broadening. The adsorption bands at 1730, 1620, 1370, 1220, and 1040 cm$^{-1}$ are assigned to C=O stretching, sp$^2$-hybridized C=C and O-H bending, C-OH stretching, C-O-C stretching and C-O vibrations of epoxy/alkoxy groups, respectively[31]. In contrast,

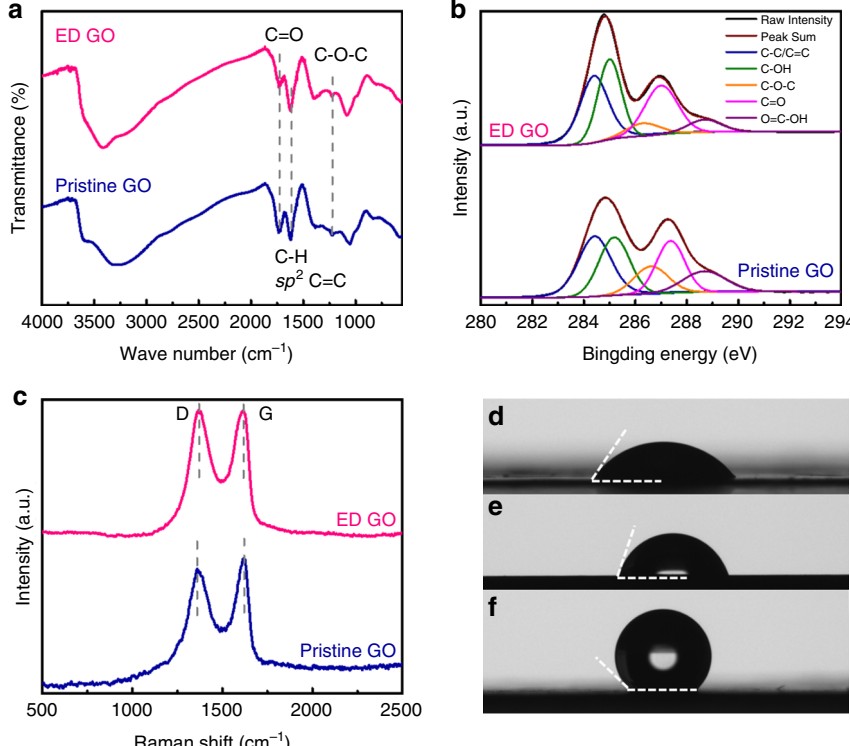

**Fig. 4** Chemical and surface properties of pristine GO and ED-GO. The pristine GO and ED-GO samples measured by **a** FTIR spectroscopy, **b** XPS and **c** Raman spectroscopy; the static WCA of **d** GO membrane prepared by filtration, **e** ED-GO membrane and **f** graphene membrane prepared by filtration

the C-OH peak of ED-GO was shifted to 3480 cm$^{-1}$, indicating the decrease of intercalated H$_2$O[24]. In addition, the intensity of carboxyl groups at 1730 cm$^{-1}$ was significantly decreased and the band of epoxy groups at 1220 cm$^{-1}$ became weaker, which proves that GO was partially reduced through preferential consumption of carboxyl and epoxy groups in the ED process. In agreement with that, the intensity of sp$^2$ carbon at 1620 cm$^{-1}$ was slightly increased for the ED-GO, implying the repair of the C = C network structure of GO[25].

XPS analyses suggest that the C/O ratio in ED-GO was increased to 2.7 from 2.3 of pristine GO due to the loss of oxygen groups (Supplementary Fig. 8), which agrees with the results of XRD and FTIR. The C1s spectra of both samples can be deconvoluted into five Gaussian peaks at 284.4 ± 0.1 eV (-C = C-/-C-C-), 285.0 ± 0.2 eV (-C-OH), 286.4 ± 0.2 eV (-C-O-C-), 287.2 ± 0.2 eV (-C = O), and 288.5 ± 0.2 eV (-O-C = O) (Fig. 4b)[32]. The -O-C = O and -C-O-C fractions were decreased from 12.2 and 12.9 at.% for pristine GO to 6.4 and 6.5 at.% for ED-GO, respectively (Supplementary Table 1), which further confirms that epoxy and carboxyl groups were partially removed.

The structural regularity of GO was analyzed by Raman spectroscopy (Fig. 4c). The spectrum of pristine GO contains a G band at 1616 cm$^{-1}$, arising from the first-order scattering of sp$^2$ carbon atoms in a 2D hexagonal lattice, and a D band at 1360 cm$^{-1}$, ascribed to the vibrations of carbon atoms in plane terminations of disordered graphite[33]. Compared to the pristine GO, the D band intensity of ED-GO is increased slightly while that of the G band is almost unchanged. The enhanced D-band suggests an imperfect repairing of the sp$^2$ bonding during reduction. On the other hand, it is known that the G band intensity will increase with the reduction of GO due to the recovery of the hexagonal carbon network[23]. Thus the unchanged G band intensity of ED-GO indicates that the reduction degree is low. The intensity ratio of the D and G band ($I_D/I_G$), which is

sensitive to the level of disorder on the basal plane of GO and defects on the carbon backbone, was slightly increased from 0.95 of pristine GO to 0.99 of ED-GO. It indicates that defects may be generated during the repair of the ordered sp$^2$ carbon network structure[25]. Yet, the small difference of $I_D/I_G$ between pristine and ED-GO also suggests a relatively low number of new defects generated during the ED process. Furthermore, the D- and G band positions of ED-GO were slightly shifted to 1367 and 1614 cm$^{-1}$ because the partial loss of epoxy and carboxyl groups weakens the electron withdrawing ability[34].

The surface wettability of ED-GO, pristine GO and graphene membranes was determined by the water contact angle (WCA). Due to the highly hydrophilic nature, the static WCA of the pristine GO membrane is 46.7 ± 1.1° (Fig. 4d). For lack of hydrophilic groups, as comparison, the WCA of the graphene membrane is 140.1 ± 1.1° (Fig. 4f). The ED-GO membrane yields a higher WCA, 71.7 ± 1.4°, than that of pristine GO, suggesting that the ED-GO membrane was still somewhat hydrophilic owing to the weak reduction (Fig. 4e). The dynamic WCA of these samples further indicate that water molecules can slightly access the bulk of pristine GO and ED-GO membranes (Supplementary Fig. 9). The surface charge of pristine and ED-GO was measured by zeta potential analysis (Supplementary Fig. 10). It reveals that both samples are negatively charged and the electronegativity is gradually enhanced in the range of pH 3–11, which is mainly contributed by the deprotonation of the carboxyl group at the edges of GO flakes[35]. But the ED-GO is less negative owing to the partial consumption of carboxyl groups during the reduction of GO, in line with the results of IR and XPS results in Fig. 4.

**Gas separation with the ED-GO@PSSHF membrane**. The permeation performance of the ED-GO@PSSHF membrane for small gas molecules and light hydrocarbons was firstly measured using a single gas feeding method (Supplementary Fig. 11). As

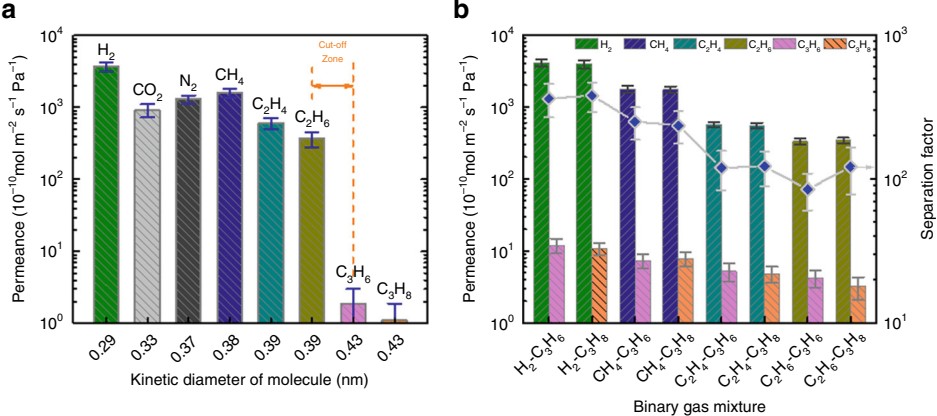

**Fig. 5** Gas permeation performance of ED-GO@PSSHF composite membrane. **a** Single-gas measurements of small gas molecules and light hydrocarbons permeation through ED-GO@PSSHF membrane ($\Delta P = 2$ bar and room temperature) and **b** separation performance of ED-GO@PSSHF membrane for binary gas mixtures (The volume ratio of mixture is 1:1, $\Delta P = 2$ bar and room temperature); error bars derived from SD

**Table 1 The ideal selectivity and separation factors of small gas molecules and light hydrocarbons over the ED-GO@PSSHF membrane ($\Delta P = 2$ bar, room temperature)**

| Ideal Selectivity (row/col.) | $H_2$ (0.29 nm) | $CO_2$ (0.33 nm) | $N_2$ (0.37 nm) | $CH_4$ (0.38 nm) | $C_2H_4$ (0.39 nm) | $C_2H_6$ (0.39 nm) | $C_3H_6$ (0.43 nm) |
|---|---|---|---|---|---|---|---|
| $CO_2$ (0.33 nm)[a] | 4.1 | | | | | | |
| $N_2$ (0.37 nm) | 2.9 | 0.7 | | | | | |
| $CH_4$ (0.38 nm) | 2.3 | 0.6 | 0.8 | | | | |
| $C_2H_4$ (0.39 nm) | 6.1 | 1.5 | 2.1 | 2.6 | | | |
| $C_2H_6$ (0.39 nm) | 10.1 | 2.5 | 3.6 | 4.3 | 1.7 | | |
| $C_3H_6$ (0.43 nm) | 1949.2 (361.5)[b] | 478.9 | 683.8 | 836.0 (249.7)[b] | 319.1 (119.6)[b] | 192.3 (84.6)[b] | |
| $C_3H_8$ (0.43 nm) | 3366.8 (378.7)[b] | 827.2 | 1181.1 | 1443.9 (234.7)[b] | 551.1 (121.8)[b] | 332.2 (121.2)[b] | 1.7 |

[a]The kinetic diameter of the responding molecules
[b]The separation factors obtained from the binary gas mixture permeation

shown in Fig. 5a, generally, the permeances of gases with small kinetic molecular diameters (from 0.29 to 0.39 nm) are close, leading to a low ideal selectivity follows in the range of 0.6 to 10.1 for any couple comprised of $H_2$, $CO_2$, $N_2$, $CH_4$, $C_2H_4$ and $C_2H_6$ (Table 1). Neglecting the impact of membrane defects, it suggests that these gases can pass through the membrane easily and the gas permeation is dominated by Knudsen diffusion rather than molecular sieving. However, a sharp decrease of permeation rate is observed when switching from ethane to propene, which leads to a clear cutoff boundary between C2 and C3. Except for C3 hydrocarbons, the permeances are around the order of $10^{-7}$ mol m$^{-2}$ s$^{-1}$ Pa$^{-1}$, whereas propene and propane have much lower permeances on the order of $10^{-10}$ mol m$^{-2}$ s$^{-1}$ Pa$^{-1}$. Therefore, the ideal selectivity of C2/C3 hydrocarbons amounted to $C_2H_4/C_3H_{8} = 551$, $C_2H_4/C_3H_{6} = 319$, $C_2H_6/C_3H_{8} = 332$ and $C_2H_6/C_3H_{6} = 192$, respectively (Table 1), which are two orders of magnitude higher than their Knudsen selectivities (~1.2). The high ideal selectivity of C2/C3 first implies that the ED-GO@PSSHF membrane is nearly perfect-stacked with almost no defects. Importantly, it demonstrates a precise molecular sieving effect by excluding gas molecules larger than ethane (0.39 nm). It is worth noting that the 2D channel of the ED-GO layer determined by XRD, $0.36 \pm 0.03$ nm, is slightly smaller than

the molecular cutoff point presented in Fig. 5a. It is reasoned that gases with the molecular size a little bit larger than the GO 2D channel also can intercalate into the bulk of the membrane due to the flexibility of GO membranes. Such expansion of GO channels would be limited to a certain range by the uniform $\pi-\pi$ inter-action force between carbon layers. Therefore, methane (0.38 nm) and ethene/ethane (0.39 nm), which are slightly larger than the ED-GO channel, can pass through the membrane rapidly. It reveals that the flexibility of GO leads to an up-shift of the cutoff point but doesn't hinder the ED-GO@PSSHF membrane from achieving high selectivity.

The separation performance of the ED-GO@PSSHF membrane was further investigated using binary gas mixtures (50 : 50 v.%) that are composed of one small gas ($H_2$, $CH_4$, or C2 hydrocarbon) and a C3 hydrocarbon. As shown in Fig. 5b, the permeance of $H_2$, $CH_4$, $C_2H_4$, and $C_2H_6$ in the mixed-gas systems are similar to that of single gas within 15% variations. On the other hand, the permeance of C3 hydrocarbons from the binary mixtures was increased in comparison with those following from single-gas measurements. For example, the permeances of propene and propane from $H_2$-C3 mixtures were increased several times to 12.0 and10.8×$10^{-10}$ mol m$^{-2}$ s$^{-1}$ Pa$^{-1}$, respectively. In gas separation with membrane, concentration polarization (CP) results in the

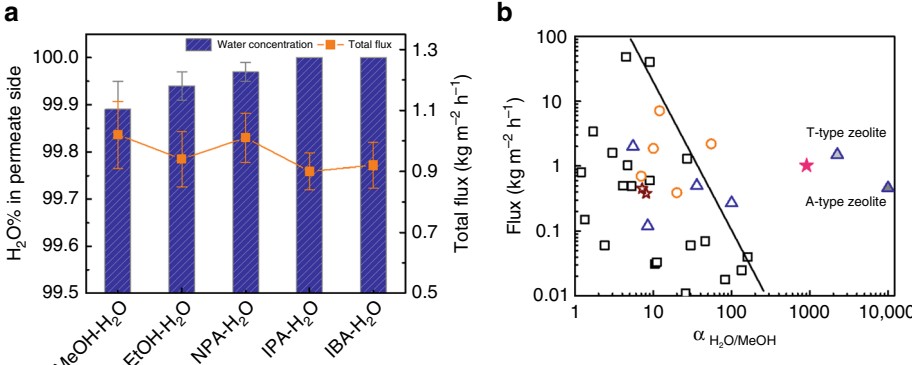

**Fig. 6** Alcohol dehydration performance and comparison with literature. **a** 50 : 50 wt.% alcohol/water binary mixture separations by pervaporation with ED-GO@PSSHF membrane at 70 °C (error bars derived from SD); **b** comparison of ED-GO@PSSHF membrane with polymeric and inorganic membranes for the separation of methanol-water mixtures (squares, cycles, and triangles represent polymeric, silica, and zeolite membranes, respectively, open and solid stars indicate the reported GO based membranes and ED-GO@PSSHF membrane, respectively)

increase of the slower-permeating gas when the preferentially permeating gas is rapidly depleted and the slower-permeating gas is enriched near the membrane surface[36]. But the CP has been minimized here by using a high feed flux (100 mL min$^{-1}$). In addition, the permeances of $H_2$ and $CH_4$ from the mixtures were also slightly increased, which are opposite to the CP effect on faster-permeating components. Thus the CP effect could be neglected here. It's logical that the 2D channel size of ED-GO was expanded to some extent by the faster-permeating gas due to the flexibility and weak interaction between GO flakes, facilitating the access of larger molecules. As shown in Fig. 5b and Table 1, the separation factors (SFs, α) of these binary mixtures exhibit the impressive molecular sieving effect for the mixture separation.

Membrane separation of C2/C3 hydrocarbons with superior performance has been reported for a ZIF-8 membrane, for which a $\alpha_{ethane/propane}$ of 80 and a ethane permeance on the order of 10$^{-7}$ mol m$^{-2}$ s$^{-1}$ Pa$^{-1}$ were achieved from the separation of equimolar binary mixture[37]. By comparison, the ED-GO@PSSHF membrane exhibited 50% higher $\alpha_{ethane/propane}$ and a comparable permeability on the same order. It suggests that the stacked ED-GO membrane with narrowed 2D channels is able to achieve a comparable sieving performance as the regular microporous membranes. A GO@PSSHF membrane with high selectivity for C2/C3 hydrocarbon separation has a high potential to meet the emergent separation requirements in many industrial processes. For instance, the products of methanol-to-olefins (MTO) contain ~90% C2-C4 light-olefins, for which an efficient method is needed to take the main product ethylene out[38]. Similar demands also exist for the separation of Fischer-Tropsch-to-olefins (FTO) products or natural gas purification[39]. In addition, the ED-GO@PSSHF membrane further provides high separation factors of several hundred for the separation of the small gases from C3 molecules, indicating an even larger potential for separation of such mixtures (Table 1).

**Alcohol dehydration with the ED-GO@PSSHF membrane**. The ED-GO@PSSHF membrane was tested in alcohol dehydration via pervaporation using alcohol (C1–C4)—water binary mixtures (50 : 50 wt.%) at 70 °C (Supplementary Fig. 12). For large-size alcohols like *iso*-propanol (IPA) and *iso*-butanol (IBA), as shown in Fig. 6a, the alcohol concentrations on the permeate side were below the detection limit of our GC (0.01%), revealing ultrahigh selectivity towards water. In the case of ethanol–$H_2O$ binary mixture separation, the purity of $H_2O$ on the permeate side was as high as 99.94%, translating to a separation factor $\alpha_{H_2O/EtOH}$ of

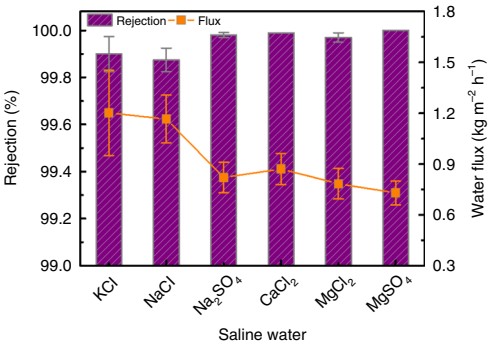

**Fig. 7** Desalination of saline water over ED-GO@PSSHF membrane. The concentration of saline aqueous solutions is 0.1 mol L$^{-1}$ and the vacuum membrane distillation temperature is 60 °C; error bars derived from SD

1665, which is one order magnitude higher than that of the typical filtration-prepared GO membranes[5,7] and even comparable with that of NaA zeolite membranes[40]. Interestingly, very high water purity, 99.89%, was also achieved for the methanol-water mixture, yielding a $\alpha_{H_2O/MeOH}$ of 908.

It is known that a water molecular layer is easily intercalated into the GO lamellas and connected with GO through hydrogen bonding in humid environments due to the inherent hydrophilicity of GO, which results in an increase of the distance between GO lamellas[41]. Thus the GO based membranes assume different states in the "dry" gas separation and "wet" aqueous phase permeation. The XRD patterns of pristine GO implies that the $d$-value increased from 0.80 nm in the dry state to 1.15 nm following water adsorption (Fig. 3a and Supplementary Fig. 13), in line with the literature[2,42]. Similarly, the ED-GO also exhibits a $d$-value growth from 0.73 to 0.96 nm upon water incorporation. The loss of oxygen-containing groups in the ED-GO may weaken the capacity for water capture, leading to a smaller expansion of the $d$-value ($\Delta = 0.23$ nm) than that of pristine GO ($\Delta = 0.35$ nm) after water uptake. Considering the thickness of graphite/rGO (0.34–0.40 nm), therefore, the 2D channels of the ED-GO and pristine GO in water are estimated at 0.59 ± 0.03 and 0.78 ± 0.03 nm, respectively.

On the other hand, the effective size of methanol, ethanol and propanol in water are ~0.57, ~0.68, and ~0.90 nm, respectively, based on their hydrated radii[2,43]. Obviously, it can be readily understood that C2 or larger alcohols could be rejected by the ED-GO@PSSHF membrane because these hydrated alcohols are much bigger than the 2D channel of wet ED-GO@PSSHF

membranes. In the case of methanol-water mixture, the size of hydrated methanol is very close to the 2D channel of ED-GO. Therefore, the steric hindrance effect caused by membrane channels became serious, which impacts the access and diffusion of hydrated methanol. In contrast, the water component possesses both higher solubility and diffusivity because the hydrophilic membrane trends to capture and hold more water and, moreover, the larger ratio of channel size to water molecular size means lower diffusion resistance. Based on the solution-diffusion model, both diffusivity and solubility contribute to the permeability of a species. Thus the high separation factor between water and methanol results from the good solubility and diffusivity of water in ED-GO layers. The water fluxes of the ED-GO@PSSHF membrane ranged from 0.90 to 1.05 kg m$^{-2}$ h$^{-1}$, which are on the same order as the GO membranes prepared by filtration[5, 6].

As methanol is the smallest alcohol, the methanol dehydration performance of ED-GO@PSSHF membrane was compared with those of polymeric, inorganic and GO based membranes (Fig. 6b)[8, 44, 45]. For polymeric membranes, the separation factors decrease as the liquid fluxes increase, forming an upper bound depends on the flux and separation factor. Our ED-GO@PSSHF membrane resides far above the upper bound for polymeric membranes and also shows superior performance compared with silica membranes. The ED-GO@PSSHF membrane is even comparable with A-type and T-type zeolite membranes, which demonstrates the extraordinary molecular sieving effect. Solvent dehydration is a common purification process in industry, which is normally costly and energy intensive, especially for the (near-) azeotropic systems. The ED-GO@PSSHF membrane can sieve the smallest alcohol out, providing a versatile method for solvent dehydration.

**Desalination with the ED-GO@PSSHF membrane**. Seawater desalination with membrane has been gaining more and more importance as a means of clean water supply. Simulations indicate that the pore size of membrane should be smaller than 1.0 nm to achieve high salt rejection[46]. Our ED-GO@PSSHF membrane readily meets this requirement due to the tight 2D channels. Therefore, desalination with the ED-GO@PSSHF membrane was tested using various saline solutions (0.1 mol salt L$^{-1}$) by the vacuum membrane distillation (VMD) method at 60 °C. As shown in Fig. 7 and Supplementary Table 2, almost 100% salt rejections were achieved for the saline solutions of KCl (99.90%), NaCl (99.88%), Na$_2$SO$_4$ (99.98%), CaCl$_2$ (99.99%), MgCl$_2$ (99.97%), and MgSO$_4$ (100%). In the desalination, cations and anions permeate through a membrane in stoichiometric amounts to maintain charge neutrality, which means that the permeation of ions is dominated by the larger ion of the cation-anion couple. On the basis of the hydrated ion radius, the effective size of ions increases in the order of 0.66 (K$^+$), 0.66 (Cl$^-$), 0.72 (Na$^+$), 0.76 (SO$_4^{2-}$), 0.82 (Ca$^{2+}$), and 0.86 nm (Mg$^{2+}$)[2, 47]. According to the sieving mechanism, even the smallest K$^+$ and Cl$^-$ are much larger than the 2D channel size of the ED-GO membrane in water (0.59 ± 0.03 nm), in line with their nearly complete rejections. In contrast, the typical GO membranes without channel size control have incomplete ion rejections for small ions due to the larger 2D channels between 0.74 and 0.95 nm[2, 11, 12].

Apart from the size sieving, the electrostatic interaction between ions and membrane fixed charges may also contribute to the ion rejection due to the electronegativity of the ED-GO membrane. This non-sieving mechanism is known as Donnan exclusion effect[48]. The charged membrane tends to reject co-ions for the repulsion and the requirement of a stoichiometric counter-ion balance to maintain electric neutrality of the solution results in the rejection of the whole salt. For the pristine GO@PSSHF membrane prepared by pressure-filtration, the rejection of the salt solutions follows the order of Na$_2$SO$_4$ > MgSO$_4$ > NaCl ≈ MgCl$_2$

(Supplementary Fig. 14), which is generally consistent with the Donnan exclusion dominated GO membranes[18, 48]. In contrast, the rejections of these four salts by the ED-GO@PSSHF membrane mainly depend on the effective ionic size rather than the charge ratio of salt (Fig. 7), which means the steric hindrance or sieving effect has become a governing factor in the desalination with the ED-GO@PSSHF membrane. But it is plausible that the electrostatic interaction between the ED-GO and co-ions is still contribute to the ion rejections since the repulsion enlarging the diffusion barrier for the salt at the surface of the ED-GO membrane.

The water fluxes for different solutions were 0.73–1.20 kg m$^{-2}$ h$^{-1}$, which is similar to that of alcohol dehydration. The fluxes of NaCl and KCl solutions are higher than those in the presence of the divalent ions since the divalent ions reduce the water flux more than monovalent ions[49]. On the other hand, the water fluxes depend on the thickness of the ED-GO@PSSHF membranes (Supplementary Fig. 15), in which higher water flux was observed from the thinner membrane owing to the lower mass transfer resistance in membrane. However, the salt rejection was significantly lower when the thickness of the ED-GO@PSSHF membrane was reduced to ~30 nm. This can be ascribed to two aspects. Defects may exist since the deposited layer being too thin to fully cover the pores of PSSHF. Furthermore, the relative lower reduction during the shorter ED process results in larger 2D channels, which is consistent with the time profile analysis of the C/O ratio in Supplementary Fig. 5.

## Discussion

Apart from selectivity and permeability, stability is another key property for practical membrane applications. The mechanical stability is a critical issue for the GO based membranes due to the weak interaction between stacked GO flakes. The ED-GO@PSSHF membrane exhibited an enhanced stability through tests in various mediums including gas mixtures, alcohol aqueous solutions and saline water with different temperature (up to 70 °C) and pressure (up to 2 bar). To further confirm the stability of ED-GO@PSSHF membrane, a long-term separation test was carried out using an ethanol-H$_2$O binary mixture. The water flux of the membrane remained stable at 0.81 ± 0.05 kg m$^{-2}$ h$^{-1}$ with a 9% drop from start to the end, while the separation factor gradually increased from 1550 to 1750 simultaneously (Supplementary Fig. 16). It confirms that the membrane remained stable in water and ethanol without defect generation. The improved mechanical stability of the ED-GO@PSSHF membrane can be ascribed to the following aspects. Firstly, the rigid PSSHF substrate endows the soft GO layer with high mechanical strength. Furthermore, a stronger interaction was established between the interface of the GO layer and the substrate due to the formation of metal hydroxides on the PSSHF surface during the ED process[50]. For the bulk of the GO layer, the narrowed interspacing strengthens the π−π interaction force. Moreover, the decrease of oxygen-containing groups weakens the solubility of ED-GO in water and consequently improves the stability in aqueous media.

Based on the ED method, on the other hand, tighter ED-GO@PSSHF membranes could be obtained with deeper electro-reduction. Accordingly, the sieving points for gas molecules down-shifted to smaller scales. For example, the sieving span shifted from > 0.43 nm for a pristine GO@PSSHF membrane to 0.41 ± 0.02, 0.35 ± 0.02 and 0.31 ± 0.02 nm for ED-GO@PSSHF membranes (Supplementary Fig. 17). However, the deeply reduced ED-GO membrane also exhibits enhanced mass diffusion resistance. In this way, for example, the membrane with 0.31 ± 0.02 nm cutoff point is nearly gas tight because that the permeance of H$_2$ was reduced by two orders of magnitude and the larger gases were even undetectable.

In conclusion, we have developed a facile and practicable ED method to realize the fast deposition of ultrathin, defect-free and robust GO layers with narrowed 2D channel on PSSHF. In this way, the oxygen-containing groups were selectively electrochemically reduced, which leads to the controlled decrease of the interspacing within stacked GO. The resultant ED-GO@PSSHF membranes feature a sharp cutoff point between C2 and C3 hydrocarbons and nearly complete rejections for the smallest alcohol and ions. This demonstrates a high versatility for the separation of various types of mixtures covering gases, solvent solutions and saline water. At the same time, robust mechanical strength of the ED-GO@PSSHF membranes were also achieved due to the enhanced interactions at GO/support and GO/GO interfaces.

## Methods

**Membrane preparation**. GO powder was supplied by Cheaptubes Co. (US) and used without further purification. Homemade PSSHF with I.D./O.D. = ~1.0/1.8 mm were cut to ~65 mm and polished, washed and dried before GO deposition. The pH value and conductivity of GO suspensions ($1\,mg\,mL^{-1}$) were 3.0 and $245\,\mu S\,cm^{-1}$, respectively.

The reduction properties of GO in electro field was tested by cyclic voltammetry (CV) with an electrochemical station. $50\,\mu L$ GO suspension ($1\,mg\,mL^{-1}$) was dropped on the glass-carbon electrode (GCE) and used as a working electrode after dried in vacuum. Pt wire, Ag/AgCl (saturated KCl) and GO suspension ($1\,mg\,mL^{-1}$) were employed as counter electrode, reference electrode and electrolyte, respectively. The CV test was carried out 100 cycles under a voltage of $-1.1$ to $0.7\,V$ with a scan rate of $50\,mV\,s^{-1}$.

The time dependence of voltage in the cyclic electric field electrodeposition of GO was tested by a three-electrode system. Tubular stainless steel GO container (I. D. = 20 mm), Ag/AgCl (saturated KCl) and PSSHF were employed as counter electrode, reference electrode and working electrode, respectively. The concentration of GO suspension is $1\,mg\,mL^{-1}$. The voltage loaded on the working electrode was fixed on 3.2 V and the test time is 100 s.

For the ED-GO@PSSHF membrane preparation, the electrolytic setup consisted of an electrochemical station (Bio-Logic, VMP3), Teflon supported stainless steel tube and PSSHF (Fig. 1a and Supplementary Fig. 2). Typically, electrophoresis deposition of GO was conducted in a concentric two electrodes system, where PSSHF was used as working electrode and the stainless steel tube was set as counter electrode. The as-prepared membranes were dried overnight at 40 °C in vacuum and then stored in drying box before use. The deposition behaviors of GO on PSSHF were investigated with different $V_{WE}$–$V_{CE}$ (4.5–12.0 V), deposition time (3–150 s) and electrode spacing (4–15 mm).

**Permeation**. For separation, the membrane was fixed into the module with two ends seals by epoxy glue. The feed was introduced from the shell side of module. To avoid the effect of moisture on membrane, the feed gas was dehydrated by passing through a molecular sieve column. In the single gas permeation, the feed flow rate of gas is $20\,mL\,min^{-1}$ and the pressure of feed side is 2 bar. One end of permeate side was closed and the left end was connected to atmosphere without sweep gas. The permeate flux was measured by soap bubble flow meters (Supplementary Fig. 11). Before flux measurement, the system was kept at the desired pressure for ca. 5 min. The permeate flux of each gas is measured at least three times. For the binary mixture separation test, two types of gas are well mixed in the gas mixer and then fed into the membrane side with a constant flow rate of $50\,mL\,min^{-1}$ for each gas and the total pressure of feed side is kept at 2 bar. In the permeate side, nitrogen was introduced from one end of membrane with a constant flow rate and used as sweep gas. The component concentration in the permeate gas was analyzed by a GC (Shimadzu GC-2014C). When the feed gas was changed, the shell side of module was purged with the corresponding feeding gas firstly and the flux measurements were carried out at least 10 min later. The membrane performance was evaluated by the permeance ($J$, $mol\,m^{-2}\,s^{-1}\,Pa^{-1}$), ideal selectivity for single gas (IS) and separation factor ($\alpha$) for gas mixture, as expressed in Eqs. (1–3):

$$J = \frac{PV}{RT} \times \frac{1}{A} \times \frac{1}{t} \times \frac{1}{\Delta P} \tag{1}$$

$$IS = \frac{J_{fast\,gas}}{J_{slow\,gas}} \tag{2}$$

$$\alpha = \frac{C_{fast\,component}}{C_{slow\,component}} \times \frac{C_{0\,slow\,component}}{C_{0\,fast\,component}} \tag{3}$$

Where $P$ (Pa) and $T$ (K) are the pressure and temperature of permeate side, $R$ is gas constant, $A$ is the membrane area (~$2.5\times10^{-4}\,m^2$), $t$ (s) is the permeation time,

$\Delta P$ (Pa) is the pressure difference and $C_i$ and $C_{0i}$ are the component concentration in the permeate and feed side, respectively.

For the alcohol dehydration, pervaporation (PV) was carried out at 70 °C (Supplementary Fig. 12). The permeate side of module was connected in sequence with sample collector, which is immersed in liquid nitrogen cold trap, and vacuum pump, by which the pressure of permeate side was kept around 100 Pa. Permeation liquid was begun to collect after 1 h running of PV system. The flux was calculated from the weight difference. The composition of permeation liquid was analyzed by a GC. The membrane performance was evaluated by the separation factor ($\alpha$) and flux ($F$, $kg\,m^{-2}\,h^{-1}$) determined as Eqs. (3) and (4):

$$F = \frac{W}{A \times t} \tag{4}$$

Where $W$ is the weight of permeation (kg), $A$ is the membrane area ($m^2$), $t$ is the permeation time (h).

For the desalination, the vacuum membrane distillation (VMD) was conducted at 60 °C with $C_0 = 0.1\,mol\,L^{-1}$ salt solutions as feedstock using the same setup of alcohol pervaporation. The ion concentration ($c_i$, $mol\,L^{-1}$) was determined by inductively coupled plasma (ICP, PerkinElmer Optima 8000) and confirmed by ion chromatography (IC, Dionex ICS-3000) as needed. Therefore, the water flux ($F$, $kg\,m^{-2}\,h^{-1}$) and ion rejection ($R$, %) are calculated by Eqs. (4) and (5):

$$R = \left(1 - \frac{C_i}{C_0}\right) \times 100\% \tag{5}$$

**Characterization**. The morphology and structure of membranes was characterized by SEM (Zeiss SUPRA 55 SAPPHIRE), TEM (JEM-2100), AFM (Oxford Instrument MFP-3Dinfinity and Bruker Co Dimension Icon) and XRD (Rigaku Ultima IV). The surface chemistry of the membranes was analyzed by XPS (Thermo Fisher, K-Alpha) and FTIR (Nicolet 6700). The carbon state was determined by Raman spectroscopy (Chameleon He–Ne laser generator with $\lambda$=531.6 nm). The water contact angle was measured by a contact angle tester (OCA20, Dataphysics). Surface charging behaviors in terms of zeta potential of GO materials were determined by an electrokinetic analyzer (SurPASS, Anton Paar GmbH, Austria).

**Data availability**. All data needed to evaluate the conclusions in the paper are present in the main text, the Supplementary Information and the Supplementary Data 1-13. Additional data are available from the authors on reasonable request.

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

## Acknowledgements

We acknowledge supports from Natural Science Foundation of China (21506243) and the Youth Innovation Promotion Association of CAS (Y624211401). The authors thank Prof. Andreas Goldbach (DICP), Prof. Zhi Liu (ShanghaiTech), Dr. Xiaopeng Li (SARI) and Dr. Deng Hu (SARI) for fruitful discussions and Mrs Xinyan Wang (ShanghaiTech) for AFM assistance.

## Author contributions

The original idea was conceived by G.Z.; the experimental design and data analysis were performed by B.Q., X.H., G.Z. and W.C.; G.H.L. and G.J.L. performed the XPS and SEM measurements. The manuscript was drafted by B.Q., X.H., G.Z., Y.P., Y.Z. and Y.S.; all authors have approved to the final version of the manuscript.

## Additional information

**Competing interests:** The authors declare no competing financial interests.

