## [Peer Review File · Nature Communications]

Reviewer #1 (Remarks to the Author)

The authors reported the application of graphene membranes for the separation of C₂/C₃ hydrocarbon mixtures. The separation factor is more than 100, and the permeance is in the range of 10⁻⁷ mol/m² s pa. These are excellent results. Although graphene membranes have been studied intensively, but most of them showed only moderate separation performance for CO₂/H₂ system. This report is the first to extend the application to valuable hydrocarbon mixtures. I strongly support the publication of this manuscript.

Reviewer #2 (Remarks to the Author)

The authors reported an electrophoresis deposition of GO on conductive porous stainless steel hollow fiber support to form thin GO membranes with in situ weak reduction. The method is relative interesting. While the results and process are too premature to be published.

1. Please provide separation data for gas mixtures, since it is very important for the application as mentioned in the manuscript.
2. The ionic separation is not reliable, since the salt concentration is 0.1 M, while the pressure is only 2 bars, which will generate high osmosis pressure.
3. How long did the ionic separation test? Because the author only provided the rejection of cations, how about the anions? The zeta potential is also needed. The surface should be negatively charged since the reduction is very weak. So, if the process is short and at high concentration, electrostatic interaction may affect the rejection?
4. The term electrodeposition is better replaced by electrophoresis deposition.
5. Typically, reduction will generate more defects and wrinkles which will affect the performance. Neutron scattering can help to more detail study the channel size.
6. How about the containing water in the sample affect the separation performance, especially for gas molecules?
7. How about separation performance for membranes with different thickness and reduction degrees.
8. During the deposition, the most down part GO should be reduced more than the upper part. Does this contribute to the separation performance?
9. Please provide the integration of ID/IG from Raman spectra to discuss the reduction and defects.
10. The reduction degree effect on the separation performance is not mentioned. Only one sample was used, which is not consistent with that the author claimed that the channel size could be tuned by the process.
11. From Figure S5, it is clear that the membrane in ethanol is less stable than that in water.
12. How about the pressure effect on the separation performance? It should be very important for osmosis pressure.
13. The stainless steel support will limit the application. How about other non conductive but more commercial substrates.

Reviewer #3 (Remarks to the Author)

Title: Strict molecular sieving over 2D-interspacing narrowed graphene oxide membrane electrodeposited on porous hollow fibre

Nature Communications manuscript NCOMMS-17-03651

In the studies reported in this manuscript, graphene oxide was prepared on non-planar porous stainless steel hollow fibres by using the electro-deposition method. This work may lead to a new approach for the preparation of thin 2D materials on stainless steel hollow fibers. However, the

electro-deposition method will also limit the choice of supporting materials; in other words, the supporting materials must be electrically conductive. Overall, the manuscript could be accepted for publication in Nature Communications if the following questions/concerns can be carefully answered/addressed:

1. The authors indicated that "In this approach, oxygen-containing groups of GO were selectively electrochemically reduced, leading to a controlled decrease of the interspacing between stacked GO layers." Although the authors compared the ED GO@PSSHf and pristine GO by FTIR and XPS, adjusting the reduction degree as well as investigating the relationship between reduction degree and molecular sieving performance should also be performed and reported. In Table S2, the XPS results of ED GO materials at different deposition times were reported; however, the adjustment of deposition time was not included. The author needs to provide the justification about choosing 35s as the deposition time (line 384), since there is no data of 35s in Table S2. Note that Table S3 (line 176) should be revised into Table S2. This reviewer also has the difficulty to understand the XPS results; there might be reduction of GO during the ED process, but it appeared that the reduction degree was not high. However, based on the FTIR spectra, the reduction degree seemed to be quite high. More explanations should be provided to avoid the confusions from readers.
2. Line 53, the authors indicated that "on the contrary, limited work was proposed aiming narrow the 2D interspacing". This reviewer would disagree; to the best of my knowledge, there are many published papers regarding the reduction of graphene oxide aiming to narrow the 2D interspacing via various chemical or physical methods. Additionally, did the authors consider to further reduce the ED GO@PSSHf in this work? If so, please make necessary additions; if not, please provide a justification.
3. Line 143 and 144, the interspacing between neighboring graphene oxide layers was calculated. Why did the authors deduct the thickness of single carbon layer? Is there any reference to support this? Line 147, where is the interspacing value of 4.3 Å come from? If the interspacing of GO is ~ 4.3 Å, then what is reason that they could not reject the ions with sizes much bigger than 4.3 Å?
4. Line 129, the authors indicated that the color of GO@PSSHf was darker than the typical GO one, while the typical GO could not be seen in Fig. 2b. The authors only compared the colors of ED GO@PSSHf and PSSHF. Moreover, why the color is slightly darker?
5. Line 152, the authors obtained the interspacing of GO by counting 22 GO lamellas from the TEM images. How did the authors measure/obtain the range of the interspaces? The unit is nm in Fig 3c, should it be Å? This reviewer gets confused about how to know the average interspacing from Fig 3c.
6. The authors have to explain the reason for high rejection of ions. Size exclusion might not be the only reason, because the interspacing of GO could not be smaller than the ions sizes. Please carefully read the paper of "Adv. Funct. Mater. 2013, 23, 3693–3700".

Responses to the comments of reviewers (NCOMMS-17-03651)

We truly appreciate the reviewers for their highly professional and thoughtful comments, which offer us a precious opportunity to promote this work with deeper scope. We have carefully revised the manuscript considering these suggestions. All the changes to the manuscript as well as supplementary information were marked in colour in the revision. In this response to comments, the original comments of reviewers are copied in italic and black while our responses are provided to the comments one-to-one and marked in blue.

Reviewer #1 (Remarks to the Author):

The authors reported the application of graphene membranes for the separation of C2/C3 hydrocarbon mixtures. The separation factor is more than 100, and the permeance is in the range of 10^{-7} mol/m² s pa. These are excellent results. Although graphene membranes have been studied intensively, but most of them showed only moderate separation performance for CO₂/H₂ system. This report is the first to extend the application to valuable hydrocarbon mixtures. I strongly support the publication of this manuscript.

Response:

We appreciate the strong support of Reviewer #1. It's true that this work extended the application of GO based membrane to the separation of C2/C3 hydrocarbon systems, which is highly attractive in both scientific research and practice application. In the revision, the mixture separation performance through the membrane was also provided. The membrane depicts high separation factors (>80) for the C3 containing binary mixtures and the same order of permeance as that of single gas method, which further confirm the molecular sieving effect of the ED GO/PSSHf membranes.

More details about the further improvements of this manuscript can be found in the revision.

Reviewer #2 (Remarks to the Author):

The authors reported an electrophoresis deposition of GO on conductive porous stainless steel hollow fiber support to form thin GO membranes with in situ weak reduction. The method is relatively interesting. While the results and process are too premature to be published.

Response:

We're thankful for the professional and thoughtful comments of Reviewer #2. The manuscript was carefully revised based on these suggestions. The main improvements of the revision include gas mixture separation with ED-GO membrane, pressure-driven desalination with ED-GO membrane, structure and surface texture investigation, the separation performance of ED-GO membranes with different reduction degree as well as separation mechanism explanations for gas and liquid. The details of these improvements and responses to the comments were listed below.

1. Please provide separation data for gas mixtures, since it is very important for the application as mentioned in the manuscript.

Response:

In the revision, the gas separation performance of a newly prepared ED GO@PSSH membrane was measured by both single gas method and binary gas mixture method, respectively, which is shown in the new Fig. 5. Small gas-C3 hydrocarbon equimolar binary gas mixtures, including H₂/C₃H₆, H₂/C₃H₈, CH₄/C₃H₆, CH₄/C₃H₈, C₂H₄/C₃H₆, C₂H₄/C₃H₈, C₂H₆/C₃H₆ and C₂H₆/C₃H₈, were measured at room temperature, as shown in the new Fig 5b. In general, the ED-GO membrane shows excellent separation performance in the binary mixture separation. The separation factors for C₂/C₃ are higher than 80.

The results and discussion of binary mixture have been added in the revision.

2. The ionic separation is not reliable, since the salt concentration is 0.1 M, while the pressure is only 2 bars, which will generate high osmosis pressure.

Response:

In our manuscript, the desalination performance of ED GO@PSSH membrane was measured by pervaporation method (PV) or called vacuum membrane distillation (VMD) rather than pressure driven method e.g. reverse osmosis (RO). RO is a process which allows the passage of water out of the saline solution with a mechanical pressure higher than the osmotic pressure of the feed solution. Thus, RO must rely on the creation of a suitably high osmotic gradient across the

membrane to achieve a reasonable flux, which requires a high operation pressure (e.g. *Water Research*, 2009, 43, 2317). Compared with the pressure driven process of RO, however, VMD is a non-pressure driven but thermally driven membrane process, in which vapor molecules are transported through micropores or microchannels of membranes as distillate (*J. Membr. Sci.* 2012, 415–416, 816; 2002, 197, 117; *J. Mater. Chem. A* 2015, 3, 5140). The driving force for mass transfer in the VMD process is the vapor pressure difference induced by the temperature difference across the membrane and a vacuum in the permeate side. VMD could be conducted at lower operating temperatures than conventional distillation and under lower operating pressures than pressure-driven membrane processes e.g. RO. VMD is a rising method for the treatment of high salt concentration saline water, which has been widely used for various membrane materials including GO based membranes. For example, the VMD has been employed for the desalination over PAN supported GO composite membrane (*J. Mater. Chem. A* 2015, 3, 5140).

Therefore, the ionic separation is credible with the VMD method in our manuscript, although the conditions of 0.1 M salt concentration and low pressure difference were chose in the desalination test.

At the same time, we're aware of that the oversimplified descriptions about the desalination experiments in the manuscript may lead to misunderstanding. Thus we have emphasized the experimental details in both the revised manuscript and supplementary information file.

In addition, we also realized the necessity to conduct the desalination by RO method because the comments 2, 3 and 12 are all related to the separation behaviors of pressure-driven method. To display the capacity of ED-GO composite membrane in a wider range, therefore, we attempted the desalination separation using RO method. The RO results were depicted as the supplementary Figure S16 and discussed in the revision, which are explained in detail in the following responses to the corresponding comments.

3. How long did the ionic separation test? Because the author only provided the rejection of cations, how about the anions? The zeta potential is also needed. The surface should be negatively charged since the reduction is very weak. So, if the process is short and at high concentration, electrostatic interaction may affect the rejection?

Response:

For the question “How long did the ionic separation test?”:

The sample collection time for the VMD is in the range of 8-15 h, depending on the flux of the membrane for different solutions. Before sample collection, the separation setup was run under the conditions of 60 °C, 3ml/min feed rate and vacuum for 2 h.

These experimental details were also added in the revision of supplementary information.

To the question “*Because the author only provided the rejection of cations, how about the anions?*”:

The anions used in the VMD desalination of former Figure 7 include Cl^- and SO_4^{2-} . To determine the cations rejections, therefore, these two kinds of anions were further investigated by using co-cation salts i.e. NaCl and Na_2SO_4 for the VMD desalination over a newly prepared ED GO@PSSHF membrane. The cation and anion concentrations of the collected samples were measured by ICP and ion chromatography, respectively. The results were listed in Table S2. In general, the anions rejections are almost the same as that of the corresponding cations rejections for both saline solutions. Cations and anions permeate through membrane in stoichiometric amounts to maintain charge neutrality of the solutions, therefore, the same rejections of anions / cations are consistent with the concept of membrane desalination.

We further understand that the reviewer likes to know the cations rejection in different salts with same anion. For the purpose of comparison, the concentration and temperature were kept the same as previous work in Figure 7. These separations of Na_2SO_4 and MgSO_4 were appended to the revised Figure 7. At the same time, the separations of KCl , CaCl_2 and MgCl_2 over the newly prepared membrane were also recorded, which was added to the revised Figure 7.

For the comment “*The zeta potential is also needed. The surface should be negatively charged since the reduction is very weak.*”

The zeta potential of pristine GO and ED-GO was measured for a wide pH range of 3-11 and added in Supplementary Figure S10. Both the pristine GO and ED GO membranes are negatively charged in the range of pH 3-11, which is mainly due to the deprotonation of the carboxyl group at the edges of the GO nanosheets (*Environ. Sci. Tech.*, 2015, 16, 10235). But the ED-GO is less negative owing to the partial consumption of carboxyl groups during the reduction of GO, in line with the results of IR and XPS results in Fig. 4.

The discussion was added in the revision.

For the comment “So, if the process is short and at high concentration, electrostatic interaction may affect the rejection?”:

For the membrane with fixed charges, the electrostatic interaction is considered as a non-sieving effect for the desalination, which is known as Donnan exclusion effect. Based on the Donnan exclusion effect, co-ions tend to be rejected due to their interactions with fixed electric charges. Counter-ions in binary electrolytes are transferred stoichiometrically owing to the zero electric current condition. Therefore, a salt as a whole is rejected. The Donnan exclusion is marked by a characteristic dependence of rejection on $Z_{\text{co}^{\oplus}\text{ions}}/Z_{\text{counter}^{\ominus}\text{ions}}$ (Z refers to the valence). To evaluate the role of Donnan exclusion in our membrane, we compared the desalination performance of typical (pristine) GO membrane with the ED-GO@PSSHF by using 0.1 mol/L NaCl, Na₂SO₄, MgCl₂ and MgSO₄ solutions. For the pristine GO@PSSHF, the rejection for the salt solutions follow the order of Na₂SO₄ > MgSO₄ > NaCl ≈ MgCl₂ (Supplementary Figure S14), in line with the negatively charged GO membranes being dominated by Donnan exclusion (*Environ Sci Tech*, 2015, 16, 10235; *Adv. Funct. Mater.* 2013, 23, 3693). It suggests that the electrostatic interaction between ions and membrane contributes to the salt rejection for the charged GO membrane when the steric hindrance/sieving effect is relatively weak. In contrast, the rejections of these four salts with ED-GO@PSSHF membrane are in the order of MgSO₄ > MgCl₂ ≈ Na₂SO₄ > NaCl, which normally depend on the ionic effective size rather than the ratio of $Z_{\text{co}^{\oplus}\text{ions}}/Z_{\text{counter}^{\ominus}\text{ions}}$ (new Fig. 7 in main text). It reveals that the steric hindrance or sieving effect has become a governing factor in the desalination over ED-GO@PSSHF membrane. But it is plausible that the electrostatic interaction between the ED-GO membrane and co-ions is still contribute to the ion rejections since the repulsion enlarging the diffusion barrier for the salt at the surface of the ED-GO membrane.

The corresponding discussion has been added in the revision.

4. The term *electrodeposition* is better replaced by *electrophoresis depositon*.

Response:

The term electrodeposition was replaced by electrophoresis deposition in the revision.

5. Typically, reduction will generate more defects and wrinkles which will affect the performance. Neutron scattering can help to more detail study the channel size.

Response:

It is a very nice idea since neutron scattering is a powerful technology to investigate the defects and structure of light elemental materials. Powder Neutron diffraction has been used for the investigation of graphene based materials (*Journal of Applied Crystallography*, 2015, 48, 1429; *Nanoscale*, 2014, 21, 13082). According to this suggestion of Reviewer 2, we did our best to contact with the Neutron facilities. However, it's very difficult to finish the booking and measurements within several months in China and even more time consuming going abroad due to the scarcity of this kind of instrument. Unfortunately, we did not receive any positive response with a definite schedule from the facilities. Actually, this is one main reason that the revision work costs us three months. But we will consider it for future studies when we can get beam time at a neutron reactor.

Instead, we have analyzed the microstructure (defects, channel size) of our membranes in much detail using complementary available technologies (XRD, TEM and AFM) in the revision. For the d-value of stacked GO sheets, XRD analysis is reliable and widely used to determine the phase structure and d-value of GO based membranes (e.g. *Science*, 2012, 335, 442; 2014, 343, 752; *Advanced Functional Materials*, 2015, 36, 5809; *ACS Nano*, 2013, 7, 1395). In the revision, more XRD results of the GO at different states (before and after ED, dry or in water) were provided and discussed (Fig. 3a and Figure S13). Using TEM measurements, we observed the stacked structure of GO sample. In addition, we also calculated the interplanar spacing of GO from the fast Fourier transform of TEM image. To investigate the surface texture and interspacing of stacked GO, we measured the GO bilayers using AFM (Fig. 3c-e). In summary, these methods provide supportive information of the channel size of the ED-GO membrane. Furthermore, AFM and Raman methods were employed for defects evaluation. The AFM height profile of ED-GO layer indicates a continuous surface texture without visible defects. From the Raman spectra, the small difference of I_D/I_G between pristine and ED GO also suggests a relatively low number of new defects generated in the ED process.

These results are illustrated in the new Figure 3 and Figure S13. The discussion for the channel size and defect analysis has been added in the revision.

6. How about the containing water in the sample affect the separation performance, especially for gas molecules?

Response:

Water molecules are easily adsorbed on the GO materials due to the abundant hydrophilic groups of GO. The adsorbed water molecules tend to reside in the 2D channels of GO by the hydrogen bonding interaction with functional groups of GO (*Sci Rep*, 2013, 3, 2714; *ACS Nano*, 2013, 7, 1166). The intercalated water molecular layers definitely impede the entrance and diffusion of other gas molecules. Therefore, the gas permeation through GO based membranes is highly sensitive to the environmental humidity. In the membrane area, the effects of water vapor on the gas permeation were firstly reported by A. Geim's group. Although the interspacing of GO lamellas become larger with the intercalation of water, the GO membrane still exhibited complete rejection to small gases even He (*Science*, 2012, 335, 442). It was also found that the gas permeance drop to zero when the nitrogen containing 0.02% water (*Carbon*, 2016, 106, 164).

Based on these knowledge and our experiences, the feed gases were thoroughly dehydrated by a molecular sieve column to remove water impurity. Moreover, the membranes for gas separation test were stored in a drying box or glove box before use.

For the separation of aqueous solutions over GO based membranes, obviously, water is a feeding component and also the fast-permeating component. Therefore, the water effects normally are not under consideration in aqueous solution separation.

7. How about separation performance for membranes with different thickness and reduction degrees.

Response:

The thickness effect on the separation performance of saline water was investigated using membranes with different thickness from ~30 nm to 1100 nm (Supplementary Figure S15). In general, the water fluxes depend on the thickness of the ED-GO@PSSHf membranes, in which higher water flux was observed from the thinner membrane owing to the lower mass transfer resistance in membrane. However, the salt rejection was significantly lower when the thickness of the ED-GO@PSSHf membrane was reduced to ~30 nm. This can be ascribed to two aspects. Defects may exist since the deposited layer being too thin to fully cover the pores or large defects of PSSHF. Furthermore, the relative lower reduction during the shorter ED process results in

larger 2D channels, which is consistent with the time profile analysis of the C/O ratio in Figure S5.

The gas permeation performance over the ED-GO membranes with different reduction degrees was investigated in the revision. Generally, tighter ED-GO@PSSHF membranes could be obtained with deeper electro-reduction. Accordingly, the sieving points for gas molecules down-shifted to smaller scales. For example, the sieving span shifted from > 0.43 nm for a pristine GO@PSSHF membrane to 0.41 ± 0.02 , 0.35 ± 0.02 and 0.31 ± 0.02 nm for ED-GO@PSSHF membranes with increased reduction degrees (Supplementary Figure S18). On the other hand, the deeply reduced ED-GO membrane also exhibits enhanced mass diffusion resistance. In this way, for example, the membrane with ~ 0.31 nm cutoff point is nearly gas tight because that the permeance of H_2 was reduced by two orders of magnitude in comparison with that of pristine GO@PSSHF and the larger gases were even undetectable.

Both thickness and reduction effect on the performance of membrane were discussed in the revised manuscript and supplementary file.

8. During the deposition, the most down part GO should be reduced more than the upper part. Does this contribute to the separation performance?

Response:

The most down part GO was less reduced than the upper part. In the ED system, GO was reduced on the counter electrode firstly and then driven by the electrostatic force to stack onto the surface of the working electrode (*ACS Nano*, 2009, 9, 2653). In our ED process, the membrane support i.e. the porous stainless steel hollow fiber is set as the working electrode (WE) and the stainless steel GO container is the counter electrode (CE). Therefore, GO is reduced by stainless steel container. In the manuscript, this process was analyzed and discussed. Fig. 1c shows a reduction signal at -0.77 V which can be assigned to the irreversible reduction of GO, indicating that GO could be reduced if V_{CE} is more negative than that value. It was also found that V_{CE} was more negative than -0.77 V when $V_{WE}-V_{CE}$ was varied in the range of 4.3-5.3 V (Fig. 1d). It indicates that GO reduction occurred when the DC voltage of $V_{WE}-V_{CE}$ is set higher than 4.3 V. In this work, the DC voltages of $V_{WE}-V_{CE}$ used for membrane preparation are higher than 4.3 V to ensure the reduction of GO suspension on the counter electrode.

In the revision, a depth profile of C/O from the GO/support interface to the membrane top surface of a ED-GO@PSSHf membrane was recorded, as shown in Figure S5b, which reveals that the C/O increased from the GO/PSSHf interface and then remained relatively stable through the membrane bulk and to the top surface. It proves that the most down part of GO was less reduced than the upper parts.

As shown in Figure S18, the ED-GO membranes display a smaller molecular cutoff point when the reduction degree of GO is higher. It indicates that the sieving performance of ED-GO membrane is mainly dominated by the deeper reduced parts of the membrane. Thus it's reasonable that the upper part GO layers contribute more for the molecular sieving effect due to tighter structure while the most down part GO contribute more for the diffusion owing to low diffusion resistance for the intercalated molecules/species in bigger 2D channels.

The depth profile of C/O ratio of the ED GO membrane was discussed in the revision.

9. Please provide the integration of I_D/I_G from Raman spectra to discuss the reduction and defects.

Response:

The discussion for the intensity ratio of I_D/I_G was provided in the revision as follows:

“The intensity ratio of the D and G band (I_D/I_G), which is sensitive to the level of disorder on the basal plane of GO and defects on the carbon backbone, was slightly increased from 0.95 of pristine GO to 0.99 of ED GO. It indicates that defects may be generated during the repair of the ordered sp^2 carbon atoms network structure.²³ But the small difference of I_D/I_G between pristine and ED GO also suggests a relatively low amount of new defects generated in ED process.”

10. The reduction degree effect on the separation performance is not mentioned. Only one sample was used, which is not consistent with that the author claimed that the channel size could be tuned by the process.

Response:

Based on this comment, we investigated the reduction degree effect on the gas separation for the revision. For comparison, we selected four membranes with different reduction degree. In addition, we also tested gas separation, ion separation and long term stability over newly prepared

ED-GO membranes. In the revision, the related permeation results of these membranes are shown in the main text Figs. 5&7 and supplementary Figures S15-S18.

The details of reduction effect on the gas separation can be found in the revision as explained in our response to the Comment #7.

11. From Figure S5, it is clear that the membrane in ethanol is less stable than that in water.

Response:

It's true that the optical images cannot be used to strictly prove the stability of the membrane. To further prove the membrane stability in water and solvents, we tested an ED GO@PSSHf membrane for the pervaporation separation of an ethanol-water mixture for 250 h. During the long-term separation test, both a relatively stable flux and selectivity were obtained, confirming the stability of ED-GO membrane in the media of water and ethanol (Figure S17).

The stability test result is shown in Supplementary Figure S17 and discussed in the revision.

12. How about the pressure effect on the separation performance? It should be very important for osmosis pressure.

Response:

To investigate the pressure effects on the salt separation, we conducted reverse osmosis separation experiment with different pressures. Due to the limited seal strength, the RO process was tested only using saline water with 200 ppm salt. A pressure, in the range of 0.5 to 3.0 bar, was applied to the feed side. Under 3 bar pressure, the salt rejections are in the range of 97.70 to 99.95% and mainly depended on the ion size, confirming the great sieving effect of ED-GO@PSSHf membrane on the ions (Supplementary Figure S16a). In comparison, these salt rejections are much higher than those of the reported GO based membranes (*Adv. Mater.* 2015, 27, 249; *Adv. Funct. Mater.* 2013, 23, 3693; *Acs. Appl. Mater. Inter.*, 2015, 7, 8147). The salt rejections and the water flux increase rates both slightly decreased with the increase of the applied pressure from 0.5 to 3.0 bar (Supplementary Figure S16b), which indicates that higher pressure may drive small amount of ions into bulk of membrane.

13. The stainless steel support will limit the application. How about other non conductive but more

commercial substrates.

Response:

For the non-conductive supports, like porous ceramic and polymers, pre-modification for these materials would be needed before attempting electrophoresis deposition. Non-conductive substrates can be easily rendered conducting by putting on metal layer with different industrially used processes (ELP, PVD, CVD, etc.). For the polymer substrates, in addition, it's also reasonable to improve the conductivity of substrate by doping polymer conductors or coating a polymer + metal nanoparticle mixture thin layer on surface.

On the other hand, the porous stainless steel (PSS) is the most often used support apart from the ceramic support in the inorganic membrane area due to its high cracking resistance, moderate operability at high temperature or pressure and flexibility for module construction and sealing. Comparing with the common plate or tubular configurations, the hollow fiber configuration of PSS (PSSHf) offers additional advantages of high surface-to-volume ratio and packing density, facial scale-up and affordability. On the other side, one of the key challenges of GO based membrane is the poor mechanical strength, which can be significantly improved by supporting the soft GO layer on a rigid substrate. In this respect, the combination of GO thin layer with PSSHf is a superior solution to obtain high quality membrane. In this work, the performance and advantages of the GO@PSSHf were demonstrated on wide areas of gas permeation and liquid separation.

Reviewer #3 (Remarks to the Author):

Title: Strict molecular sieving over 2D-interspersing narrowed graphene oxide membrane electrodeposited on porous hollow fibre

Nature Communications manuscript NCOMMS-17-03651

In the studies reported in this manuscript, graphene oxide was prepared on non-planar porous stainless steel hollow fibres by using the electro-deposition method. This work may lead to a new approach for the preparation of thin 2D materials on stainless steel hollow fibers. However, the electro-deposition method will also limit the choice of supporting materials; in other words, the supporting materials must be electrically conductive. Overall, the manuscript could be accepted for publication in Nature Communications if the following questions/concerns can be carefully answered/addressed:

Response:

We're thankful for the supports and expertise provided from Reviewer #3. We have carefully revised the manuscript considering the thoughtful suggestions of the reviewer. The details of our responses to the comments were listed below.

1. The authors indicated that "In this approach, oxygen-containing groups of GO were selectively electrochemically reduced, leading to a controlled decrease of the interspacing between stacked GO layers." Although the authors compared the ED GO@PSSHf and pristine GO by FTIR and XPS, adjusting the reduction degree as well as investigating the relationship between reduction degree and molecular sieving performance should also be performed and reported.

Response:

In the revision, we investigated the reduction degree effect on the molecular sieving performance of gas separation. The single gas permeation performance over the ED-GO membranes with different reduction degrees was measured. Generally, tighter ED-GO@PSSHf membranes could be obtained with deeper electro-reduction. Accordingly, the sieving points for gas molecules down-shifted to smaller scales. For example, the sieving span shifted from > 0.43 nm for a pristine GO@PSSHf membrane to 0.41 ± 0.02 , 0.35 ± 0.02 and 0.31 ± 0.02 nm for ED-GO@PSSHf membranes with increased reduction degrees (Supplementary Figure S18). On the other hand, the deeply reduced ED-GO membrane also exhibits enhanced mass diffusion resistance. In this way, for example, the membrane with ~ 0.31 nm cutoff point is nearly gas tight because that the permeance of H_2 was reduced by two orders of magnitude and the larger gases were even undetectable.

The related results and discussion can be found in revised SI and manuscript.

In Table S2, the XPS results of ED GO materials at different deposition times were reported; however, the adjustment of deposition time was not included. The author needs to provide the justification about choosing 35s as the deposition time (line 384), since there is no data of 35s in Table S2. Note that Table S3 (line 176) should be revised into Table S2.

Response:

In the revision, more details about the deposition condition optimization were provided, as shown in Figures S3-S7. The justification about deposition conditions was described in detail in the main text and supplementary information. Fig. 1c shows a reduction signal (peak III) at -0.77 V which can be assigned to the irreversible reduction of GO, indicating that GO could be reduced if V_{CE} is lower than that value. Moreover, it was found that V_{CE} was more negative than -0.77 V when $V_{WE}-V_{CE}$ was varied in the range of 4.3-5.3 V (Fig. 1d). It indicates that GO reduction occurred when the DC voltage of $V_{WE}-V_{CE}$ is set higher than 4.3 V. On the other hand, the ED-GO layer quality is difficult to control under high $V_{WE}-V_{CE}$ due to the very fast deposition rate and bubbles impact (Supplementary Figure S3). Thus the $V_{WE}-V_{CE}$ was normally set around 4.5 V. Under this electrochemical environment, the thickness of ED-GO@PSSHf is nearly linear dependent on the ED time within 90 s (Supplementary Figure S4). The C/O ratio of ED-GO layer increased fast from 2.2 to 2.6 in the beginning and then sluggishly grew with fluctuations to around 2.7 in the following. On the other hand, the deposition rate is evidently impacted by the electrode gap (Supplementary Figure S6). Therefore, taking GO reduction, bubbles impact, thickness and defects control into account, the ED-GO@PSSHf membranes for separation tests were fabricated in a 1 mg mL⁻¹ GO suspension with 9 mm electrode spacing and 4.5 V $V_{WE}-V_{CE}$ during 35 s.

It's noted that we replaced the deposition results of 5 V or 50 s in the former SI with the new results of 4.5 V and 35 s. In addition, the new XPS results of the corresponding conditions were not listed in the Tables. Instead, they were added to the figures of thickness dependence for the clarity.

All the numbers of new Tables and Figures were corrected in the revision.

This reviewer also has the difficulty to understand the XPS results; there might be reduction of GO during the ED process, but it appeared that the reduction degree was not high. However, based on the FTIR spectra, the reduction degree seemed to be quite high. More explanations should be provided to avoid the confusions from readers.

Response:

To avoid the confusions from readers, we rechecked more samples by FTIR, XRD and XPS. The clearer results were provided in new Fig. 4 in the revision. From the IR spectrum of ED-GO, the intensity of the carboxyl groups at 1730 cm^{-1} was significantly decreased and the band of epoxy groups at 1220 cm^{-1} became weaker in comparison with pristine GO, which proves that ED-GO was partially reduced through preferentially consuming carboxyl and epoxy groups in the ED process. From the XPS C1s spectra, the -O-C=O and -C-O-C contents were decreased from 12.2 and 12.9 at.% for pristine GO to 6.4 and 6.5 at.% for ED-GO, respectively.

The related discussion was added in the revision.

2. Line 53, the authors indicated that “on the contrary, limited work was proposed aiming narrow the 2D interspacing”. This reviewer would disagree; to the best of my knowledge, there are many published papers regarding the reduction of graphene oxide aiming to narrow the 2D interspacing via various chemical or physical methods. Additionally, did the authors consider to further reduce the ED GO@PSSHf in this work? If so, please make necessary additions; if not, please provide a justification.

Response:

It's true that there are many methods to obtain reduced graphene oxide from graphene oxide. To describe it more clearly, we revised it as “On the contrary, narrowing the 2D interspacing of GO based membranes is more attractive as the tighter GO membrane is actually desired for small molecules sieving.” At the same time, we also summarize reduction methods for graphene oxide as “Various reduction methods, like chemical reduction and thermal deoxygenation, have been purposed for the preparation of reduced graphene oxide.^{21, 22}”.

The membranes with different reduction degree were investigated in the revision. The details can be found in the revision as explained in our response to the part one of Comment #1.

3. Line 143 and 144, the interspacing between neighboring graphene oxide layers was calculated. Why did the authors deduct the thickness of single carbon layer? Is there any reference to support this? Line 147, where is the interspacing value of 4.3 \AA come from? If the interspacing of GO is $\sim 4.3\text{ \AA}$, then what is reason that they could not reject the ions with sizes much bigger than 4.3 \AA ?

Response:

For the questions “Why did the authors deduct the thickness of single carbon layer? Is there any reference to support this?”:

The d-value of stacked GO is the distance between the centers of the graphene sheets, which can be calculated by the Bragg equation ($2d\sin\Theta=n\lambda$) from the XRD pattern. The single layer graphene with the electronic clouds around the graphene sheet extend over a distance of $\sim 3.4\text{\AA}$ (*Science*, 2012, 335, 442). For the membrane research, therefore, the interlayer “empty” spacing (which we refer to “interspacing” in the manuscript) of graphene oxide for molecules/species diffusion can be determined by deducting the thickness of single graphene layer from the d-value as described in the SI of *Science*, 2012, 335, 442. This approach has been also used in other literature e.g. *Science*, 2014, 343, 752 (in which the graphene thickness of 3.4\AA was replaced by the thickness of reduced graphene oxide monolayer $\sim 4\text{\AA}$); *Advanced Functional Materials*, 2015, 36, 5809; *Chemical Communications*, 2015, 51, 7345 and *Chemical Society Review*, 2015, 44, 5016 ($\sim 4\text{\AA}$). In the revision, we considered both values (3.4 and 4.0\AA , i.e. $3.7\pm 0.3\text{\AA}$) of the monolayer graphene/rGO and estimated the 2D channel size of stacked GO with average of “ $x\pm 0.3\text{\AA}$ ”.

For the question “*Line 147, where is the interspacing value of 4.3\AA come from?*”:

The value 4.3\AA is a typo and should be 4.5\AA . The d-value of pristine GO determined by XRD is 7.9\AA . Therefore, the free interspacing between the GO sheets is 4.5\AA with deducting monolayer thickness of graphene ($\sim 3.4\text{\AA}$). For clarity, moreover, we use “d-value” for the crystallographic distance and “2D channel size” for the free spacing for species diffusion in the revision.

For the question “*If the interspacing of GO is $\sim 4.3\text{\AA}$, then what is reason that they could not reject the ions with sizes much bigger than 4.3\AA ?*”:

In the aqueous solution media, both the 2D channel size of GO based membranes and effective ion size are changed. For the membrane, the 2D channel size will expand to some extent due to water molecules being easily adsorbed on the hydrophilic surface of the GO membrane and then accommodate in the 2D channels as extra layer (*Science*, 2012, 335, 442). For the ions in water, the aqueous salt solution can be regarded as a pseudo-liquid mixture containing free water molecules and bulkier hydrated ions formed in solution. The effective size of ions in water is estimated by the hydrated radii. In the revision, we measured the d-value of pristine GO and ED

GO after water adsorption with XRD. The XRD pattern of water adsorbed pristine GO implies that the d-value increased from 8.0 Å at dry state to 11.5 Å (Supplementary Figure S13), in line with the literature (*Science*, 2014, 343, 752; *Acs Nano*, 2013, 7, 1395). Similarly, the wetted ED-GO also displays a d-spacing growth from 7.3 Å to 9.6 Å. Considering the thicknesses of graphite/rGO, therefore, the 2D channel of pristine GO is estimated at 7.8±0.3 Å, while that of ED-GO is 5.9±0.3 Å. Therefore, the ions with effective size like 6.6 (K⁺), 6.6 (Cl⁻), 7.2 (Na⁺) and 7.6 Å (SO₄²⁻) can pass through the GO membrane but be rejected by ED-GO membrane in the respect of sieving effects.

More discussion about the separation mechanism of gas, alcohol and salt over ED-GO membrane has been added in the revision.

4. Line 129, the authors indicated that the color of GO@PSSHf was darker than the typical GO one, while the typical GO could not be seen in Fig. 2b. The authors only compared the colors of ED GO@PSSHf and PSSHf. Moreover, why the color is slightly darker?

Response:

Our original intension was to compare the ED-GO membrane with the cited literature results (i.e. typical GO membrane prepared by physical methods like spinning coating and filtration). Therefore, no image of the typical GO membrane was provided in Fig. 2b.

In the revision, we provided an optical image of the ED-GO@PSSHf membrane and pristine GO@PSSHf prepared by filtration in supplementary information (Figure S7).

In diluted dispersions, GO appears brown due to the functional groups of GO reflecting the corresponding colour while graphene (or reduced GO) appears black owing to the weak light reflection (*Chem. Soc. Rev.*, 2010, 39, 228). The colours of ultrathin GO or rGO membranes are close to the corresponding dilute dispersions. In Figure S7, the colour of the ED-GO@PSSHf membrane is slightly darker than that of the filtrated membrane, indicating that the former membrane was mildly reduced or loss of functional groups during the deposition. Empirically, therefore, the color change between brown and black could be used for estimation of reduction before the destructive characterization.

5. Line 152, the authors obtained the interspacing of GO by counting 22 GO lamellas from the

TEM images. How did the authors measure/obtain the range of the interspaces? The unit is nm in Fig 3c, should it be Å? This reviewer gets confused about how to know the average interspacing from Fig 3c.

Response:

Based on the contrast difference along the direction of the marked line drawn in Fig. 3b, the interspacing of GO lamellas can be approximated by the gap between two neighboring peaks (Carbon, 2014, 678, 670). As illustrated in Figure R1, we measured the marked peaks and calculated the average.

The unit is nm in the former Fig. 3c (i.e. Figure R1).

In the revision, we accepted the suggestion of a TEM expert that the electronic diffraction is more accurate than the contrast difference for the planar spacing evaluation. Therefore, the Fig 3c was replaced by a simulated diffraction pattern. In the TEM measurement, the distribution density of the GO TEM sample is too low to get electronic diffraction directly. Therefore, the simulation pattern was obtained by fast Fourier transform (FFT) in the revision.

Figure R1 The schematic of the evaluation of GO interspacing based on the contrast difference of TEM image.

6. The authors have to explain the reason for high rejection of ions. Size exclusion might not be the only reason, because the interspacing of GO could not be smaller than the ions sizes. Please carefully read the paper of “*Adv. Funct. Mater.* 2013, 23, 3693–3700”.

Response:

We carefully read the paper of *Adv. Funct. Mater.* 2013, 23, 3693–3700 again and it was cited in the revision.

Apart from the size exclusion effect, the electrostatic interaction between the charged membrane and ions in solution may contribute to the ion rejection. To evaluate the role of Donnan exclusion (electrostatic interaction) for our membrane, we compared the desalination performance of a typical (pristine) GO membrane with the ED-GO@PSSHf via the vacuum membrane

distillation method using 0.1 mol/L NaCl, Na₂SO₄, MgCl₂ and MgSO₄ solutions. For the pristine GO@PSSHF, the rejection for the salt solutions follow the order of Na₂SO₄ > MgSO₄ > NaCl ≈ MgCl₂ (Supplementary Figure S14), in line with the negatively charged GO membranes dominated by Donnan exclusion (e.g. *Environ Sci Tech*, 2015, 16, 10235; *Adv. Funct. Mater.* 2013, 23, 3693). It suggests that the electrostatic interaction between ions and membrane contributes to the salt rejection for the charged GO membrane when the steric hindrance is relatively weak. In contrast, the rejections of these four salts over the ED-GO@PSSHF membrane are follow the order of MgSO₄ > MgCl₂ ≈ Na₂SO₄ > NaCl, and thus mainly depends on the ionic effective size rather than the ratio of $Z_{\text{co}^{\ominus}\text{ions}}/Z_{\text{counter}^{\oplus}\text{ions}}$ (new Fig. 7 in main text). It reveals that the steric hindrance or sieving effect has become a governing factor in the desalination with the ED-GO@PSSHF membrane. But it's reasonable to assume that the electrostatic interaction between ED-GO and co-ions is still contribute to the ion rejections since the repulsion increasing the diffusion barrier for the salt at the surface of the ED-GO membrane.

We explained the effects of 2D channel size of the ED-GO membrane and the effective size of ions in solution in the response to Comment #3.

The related discussion was added in the revision.

Reviewers' Comments:

Reviewer #2:

Remarks to the Author:

Most comments raised by the reviewers have been well addressed. Only one part for the RO content, it is better to remove this part. The reason is that: 1)The applied pressure is no more than 3 bar, much lower than the RO process. 2) Even under 3 bar, the salt rejection became worse, that means it is hard to be used for RO.

Responses to the comments of reviewers (NCOMMS-17-03651A)

Reviewer #2 (Remarks to the Author):

Most comments raised by the reviewers have been well addressed. Only one part for the RO content, it is better to remove this part. The reason is that: 1)The applied pressure is no more than 3 bar, much lower than the RO process. 2) Even under 3 bar, the salt rejection became worse, that means it is hard to be used for RO.

Response:

Based on the comment, the RO content was removed in the revision.